# Path Independent Equilibrium Models Can Better Exploit Test-Time Computation

**Cem Anil**[1]* **Ashwini Pokle**[2]* **Kaiqu Liang**[3]* **Johannes Treutlein**[1,4]
**Yuhuai Wu**[5] **Shaojie Bai**[2] **Zico Kolter**[2,6] **Roger Grosse**[1]
[1]University of Toronto and Vector Institute   [2]Carnegie Mellon University   [3]Princeton University
[4]University of California, Berkeley [5]Stanford University and Google Research   [6]Bosch Center for AI
{anilcem, rgrosse}@cs.toronto.edu   kl2471@princeton.edu
{apokle, shaojieb, zkolter}@cs.cmu.edu
yuhuai@google.com   johannestreutlein@berkeley.edu

## Abstract

Designing networks capable of attaining better performance with an increased inference budget is important to facilitate generalization to harder problem instances. Recent efforts have shown promising results in this direction by making use of depth-wise recurrent networks. We show that a broad class of architectures named *equilibrium models* display strong upwards generalization, and find that stronger performance on harder examples (which require more iterations of inference to get correct) strongly correlates with the *path independence* of the system—its tendency to converge to the same steady-state behaviour regardless of initialization, given enough computation. Experimental interventions made to promote path independence result in improved generalization on harder problem instances, while those that penalize it degrade this ability. Path independence analyses are also useful on a per-example basis: for equilibrium models that have good in-distribution performance, path independence on out-of-distribution samples strongly correlates with accuracy. Our results help explain why equilibrium models are capable of strong upwards generalization and motivates future work that harnesses path independence as a general modelling principle to facilitate scalable test-time usage.

## 1  Introduction

One of the main challenges limiting the practical applicability of modern deep learning systems is the ability to generalize outside the training distribution [Koh et al., 2021]. One particularly important type of out-of-distribution (OOD) generalization is *upwards generalization*, or the ability to generalize to more difficult problem instances than those encountered at training time [Selsam et al., 2018, Bansal et al., 2022, Schwarzschild et al., 2021b, Nye et al., 2021]. Often, good performance on more difficult instances will require a larger amount of test-time computation, so a natural question arises: how can we design neural net architectures which can reliably exploit additional test-time computation to achieve better accuracy?

Equilibrium models, a broad class of architectures whose outputs are the fixed points of learned dynamical systems, are particularly suited to meet this challenge. Closely related to weight-tied recurrent models – networks that apply the same fixed neural network module repeatedly to hidden-layer activations – equilibrium models are capable of adapting their compute budget based on the input they are given. Under what conditions, if any, can this input-dependent ability to scale-up test-time compute actually lead to upwards generalization?

---

*Equal contribution. Correspondence to anilcem@cs.toronto.edu and apokle@cs.cmu.edu.

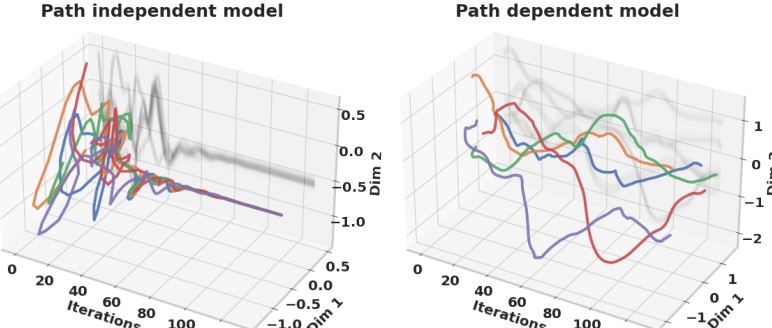

Figure 1: Trajectories of path independent models converge to the same hidden state for a given input, regardless of initialization, whereas the trajectories of path dependent models depend on initialization. Here, we display five trajectories with different initializations obtained from a path independent (left) and path dependent model (right) on the prefix-sum task, projected onto two random directions.

We argue that a key determiner of whether a learned model can exploit additional test-time computation is whether the dynamical system corresponding to the model is *path independent*; that is, whether the learned model's hidden layer activations converge to the same asymptotic behaviour (i.e. fixed point or limit cycle), regardless of the initialization of the system. For example, a simple integrator $x_{t+1} = x_t + 1$ is clearly *not* path independent, as its final state depends on the initial state $x_0$ and the number of iterations run; conversely, the system $x_{t+1} = (x_t + 1)/2$ *is* path independent, as it will converge to the solution $x_T = 1$ as $T \to \infty$ regardless of the initial condition of $x_0$. Path independence is closely related to the concept of *global stability* from control theory (see Section 9 for more).

Intuitively, path independent systems can more easily take advantage of additional test-time iterations than path dependent ones. For instance, gradient descent applied to a convex objective is path independent, and correspondingly when confronted with a more ill-conditioned problem instance, one can compensate by increasing the number of iterations. Conversely, a weather simulation is path dependent, and extending the simulation won't yield more accurate predictions of a given day's weather. Based on this intuition, we hypothesize that path independence of a learned model is a key determiner of whether it can take advantage of an increased test-time iteration budget when generalizing to harder problem instances:

> *Path Independence Hypothesis:* Models which successfully fit the training distribution with a path independent function are better able to exploit more test-time iterations to achieve higher accuracy, compared to those which fit the training distribution with a path-dependent function.

We first introduce a metric for measuring the path independence of a learned model, the *Asymptotic Alignment (AA)* score. On a wide range of tasks including sequence prediction, visual reasoning, image classification, continuous optimization and graph regression, we show that a model's AA score is strongly correlated with its performance when far more iterations are used at test time than at training time. In general, we find that path independent models increase their performance monotonically with the number of test-time iterations, whereas path dependent models degrade when the number of test time iterations exceeds those at training time. We find that input injection and implicit updates improve both the path independence and the accuracy. Furthermore, we perform an experimental manipulation by introducing regularizers which directly promote or punish path independence. We find that these manipulations, while increasing or decreasing the AA score, also have the corresponding effect on accuracy.[2]

## 2 Background

**Equilibrium Models** Equilibrium models treat computing internal representations as a fixed-point finding problem. [McClelland and Rumelhart, 1989, Liao et al., 2018, Bai et al., 2019] Concretely, letting $\boldsymbol{x} \in \mathbb{R}^{n_x}$ and $f_{\boldsymbol{w}} : \mathbb{R}^{n_x \times n_z} \mapsto \mathbb{R}^{n_z}$ stand for an input and the equilibrium model function (or

---

[2]Code will be released along with the paper

"cell") parametrized by $\boldsymbol{w} \in \mathbb{R}^{n_w}$ respectively, equilibrium models aim to solve for the fixed point $\boldsymbol{z}^* \in \mathbb{R}^{n_z}$ that satisfies $\boldsymbol{z}^* = f_{\boldsymbol{w}}(\boldsymbol{x}, \boldsymbol{z}^*)$. The cell $f_{\boldsymbol{w}}$ might represent anything from a fully connected layer to a transformer block [Vaswani et al., 2017]. We emphasize that $f_{\boldsymbol{w}}$ directly depends on the input $\boldsymbol{x}$; following existing literature, we refer to this as *input injection*. The outcome of the fixed point finding process might depend on the initial fixed point guess. To make this dependency explicit, we define the function $\text{FIX} : \mathbb{R}^{n_x \times n_z} \mapsto \mathbb{R}^{n_z}$ that maps an input $\boldsymbol{x}$ and an initial guess for the fixed point $\boldsymbol{z}_0$ to an output that satisfies the fixed point equation $\text{FIX}_{f_w}(\boldsymbol{x}, \boldsymbol{z}_0) := f_{\boldsymbol{w}}(\boldsymbol{x}, \text{FIX}_{f_w}(\boldsymbol{x}, \boldsymbol{z}_0))$. The behaviour of FIX depends on the solver that's used to find fixed points.

The most straightforward approach to solve for fixed-points is the *fixed point iteration* method, which recursively applies the function $f_{\boldsymbol{w}}$ on the internal representations $\boldsymbol{z}$ (i.e. $\boldsymbol{z}_{t+1} = f_{\boldsymbol{w}}(\boldsymbol{x}, \boldsymbol{z}_t)$). If certain conditions are satisfied (such as the fixed iterations being globally contractive – more general conditions are discussed below), this procedure converges[3] to a fixed point: $\boldsymbol{z}^* = f_{\boldsymbol{w}}(\boldsymbol{x}, \boldsymbol{z}^*)$. As solving for fixed points exactly is expensive, fixed point iterations are often terminated after a fixed number of steps or when the norm of the difference between subsequent iterates falls below a pre-selected threshold. The model weights can be updated using gradients computed via backpropagating through the full forward computational graph.

If one commits to using fixed point iterations as the root solver, then the output $\boldsymbol{z}^*$ of equilibrium models can be interpreted as the infinite-depth limit of an input-injected, weight-tied model $f_{\boldsymbol{w}}^{\infty}(\boldsymbol{x}, \boldsymbol{z}^*) = \lim_{n \to \infty} f_{\boldsymbol{w}}^{(n)}(\boldsymbol{x}, \boldsymbol{z}_0)$ where the notation $f^{(n)}$ stands for $n$ repeated applications of $f$ on its own output, and $\boldsymbol{z}_0$ stands for the fixed point initialization.

**Implicit Training of Equilibrium Models** Different training algorithms for equilibrium models can be derived by considering their implicit nature. Bai et al. [2019] solve for fixed points explicitly using black-box root finders, such as Broyden's method [Broyden, 1965] or Anderson acceleration [Anderson, 1965]. In order to avoid explicitly differentiating through the root-finding procedure, they utilize implicit differentiation to compute gradients.[4] Concretely, letting $g_{\boldsymbol{w}}(\boldsymbol{x}, \boldsymbol{z}) = f_{\boldsymbol{w}}(\boldsymbol{x}, \boldsymbol{z}) - \boldsymbol{z}$ for a fixed point $\boldsymbol{z}^*$, the Jacobian of $\boldsymbol{z}^*$ with respect to the equilibrium model weights can be given by:

$$\frac{d\boldsymbol{z}^*}{d\boldsymbol{w}} = -\left(\frac{\partial g_{\boldsymbol{w}}(\boldsymbol{x}, \boldsymbol{z}^*)}{\partial \boldsymbol{z}^*}\right)^{-1} \frac{\partial f_{\boldsymbol{w}}(\boldsymbol{x}, \boldsymbol{z}^*)}{\partial \boldsymbol{w}} \tag{1}$$

Inverting a Jacobian matrix can become computationally expensive. Recent works [Geng et al., 2021a, Fung et al., 2021] have shown that the inverse-Jacobian term in Eq. (1) can be replaced with an identity matrix i.e. Jacobian-free or an approximate inverse-Jacobian [Geng et al., 2021b] without affecting the final performance. This approximation makes the backward pass inexpensive and lightweight. Equilibrium models have been shown to achieve state-of-the-art performance on various tasks including language modelling [Bai et al., 2019], image recognition, semantic segmentation [Bai et al., 2020], object detection [Wang et al., 2020], and graph modeling [Gu et al., 2020, Liu et al., 2021a].

**Equilibrium Models vs. Depthwise Recurrent Models** Both equilibrium models and input-injected depthwise recurrent (i.e. weight-tied, fixed-depth) networks leverage weight-tying *i.e.,* they apply the same transformation at each layer, $f_{\omega}^{[i]} = f_{\omega} \, \forall i$. The two models differ in the ultimate aim of the forward pass: while depthwise recurrent models compute a (weight-tied) fixed depth computation (which may or may not approach a fixed point), the stated *goal* of equilibrium models is explicitly to find a fixed point. Weight-tied fixed depth networks by definition require backpropagation through an explicit stack of layers. Equilibrium models, however, directly solve for fixed points using (potentially black-box) solvers during the forward pass and may be trained using implicit differentiation.

**Convergence** As alluded above, in order to guarantee convergence to a unique fixed point, it suffices for the cell of the equilibrium model to be *contractive* over its input domain[5] (i.e. the singular values of its Jacobian all lie below 1). Previous work has leveraged Lipschitz constrained cells to ensure contractivity [Revay et al., 2020]. Other approaches for ensuring global convergence exist: the monotone equilibrium model architecture guarantees global convergence by utilizing an equilibrium model parametrization that bears similarities to solutions to a particular form of monotone operator

---

[3]Divergence is also a possible outcome, rendering the output of equilibrium models unusable.

[4]Bai et al. [2019] use the term *Deep Equilibrium Models* (DEQ) to refer to implicitly trained equilibrium models. To keep things more general, we categorize "explicitly trained" networks (i.e. with fixed point iterations and backpropagation) under the umbrella of equilibrium models as well.

[5]This is known as the Banach fixed-point theorem.

splitting problem [Winston and Kolter, 2020]. Unrestricted equilibrium models aren't constrained enough to guarantee convergence: they can easily express globally divergent vector fields that prohibit the existence of fixed points. It is, therefore, interesting that they can (and often) learn path independent solutions. Also note that the "infinite-depth weight tied network" interpretation of equilibrium models is less general than the implicit formulation presented above, as the latter admits unstable fixed points as well.

**Terminology and Abbreviations** We use the term "equilibrium models" to refer to the general class of networks that explicitly solve for a fixed point in the forward pass. We use the term 'solver' to refer to the use of black-box root finders like Anderson acceleration to find fixed points of an implicitly trained equilibrium model. These networks can use implicit gradients computed via implicit function theorem (IFT), Jacobian-free backward pass, or with an approximation of inverse-Jacobian. The term 'unroll' refers to equilibrium models that use regular fixed-point iterations to compute the equilibrium point. We use the abbreviation 'bp' to refer to backpropagation gradients, and 'inj' to refer to input injection. The term 'progressive net' refers to the deep thinking networks trained with progressive training as proposed by Bansal et al. [2022]. We use 'PI' and 'non-PI' to refer to path independent and path dependent networks, respectively.

## 3 Upwards Generalization with Equilibrium Models

In this section, we establish that equilibrium models are capable of strong upwards generalization. To study the effects of test time computation, it is useful to consider tasks with an explicit difficulty parameter, so that the learned models can be tested on more difficult instances which require a large number of iterations to solve correctly. We focus on multiple algorithmic generalization tasks: **prefix sum** and **mazes** by Schwarzschild et al. [2021a,b], **blurry MNIST**, **matrix inversion** and **edge copy** by Du et al. [2022]. Taken together, these tasks cover a wide range of problems from different domains, namely sequence prediction, visual reasoning, image classification, continuous optimization and graph regression. To maintain clarity and focus, we run our detailed analysis on the prefix sum and mazes tasks, and provide complementary results for the remaining tasks in the Supplementary Material (SM).

**Tasks Prefix-sum** is a sequence-to-sequence task whereby the network is given a sequence of 0-1 bits, and is trained to output, for each bit, the parity of all of the bits received since the beginning of the sequence until the current bit. We train on 10,000 unique 32-bit binary strings, and report results on binary strings of other lengths. The **mazes** task is also an image-to-image task, where the input is a three-channel RGB image. The 'start' and 'finish' positions are marked by a red and a green square respectively; walls are marked in black. The output is the optimal path in the maze that connects these two points without passing through the walls. We train on 50,000 small mazes of size $9 \times 9$, and report upward generalization results on larger mazes. Instances of each of these problems, as well as additional image classification and continuous optimization results can be found in the supplementary material. **Blurry MNIST** [Liang et al.] is a robustness-to-corruption task: one has to learn to do MNIST classification from lightly blurred images and generalize zero-shot to highly blurred ones. In the **matrix inversion task** [Du et al., 2022], the goal is to learn to invert $10 \times 10$ matrices in a way that generalizes to matrices that have worse condition number than those observed during training. **Edge copy** [Du et al., 2022] is a simple graph regression tasks that requires learning to output the input edge features, in a way that generalizes to larger graph sizes. Note that the training and test data in the latter two tasks are generated with noise added on-the-fly, as done by Du et al. [2022].

**Strong Upward Generalization** Fig. 2a shows that equilibrium models demonstrate very strong upward generalization performance compared to non weight-tied fixed-depth models. Moreover, Fig. 2b shows that increasing inference depth consistently improves performance—especially on harder problem instances.

## 4 Path Independence

Having intuitively motivated the idea of path independence in Sec. 1, we now define it formally: we say that the computation performed by a recurrent operator computing function $f_{\boldsymbol{w}}$ on an input $\boldsymbol{x}$ is path independent if it converges to the same limiting behavior regardless of the current state $\boldsymbol{z}_t$. As a special case, if the computation is convergent, this property is equivalent to the existence of a unique fixed point $\boldsymbol{z}^*$ such that $f_{\boldsymbol{w}}^{\infty}(\boldsymbol{x}, \boldsymbol{z}_0) = \boldsymbol{z}^*$ for any $\boldsymbol{z}_0$. However, our definition allows for other behaviors such as limit cycles (see Sec. 7).

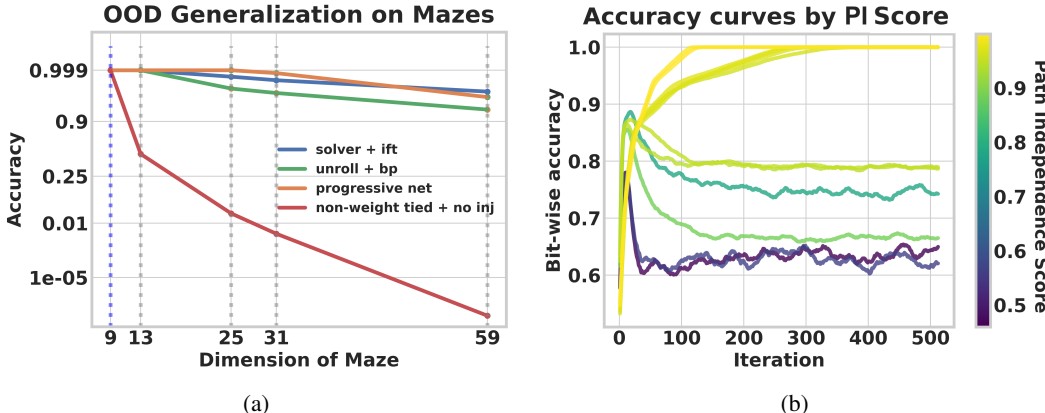

Figure 2: (**left**) Strong upward generalization on mazes by PI models. Models were trained on $9 \times 9$ sized mazes and tested for upward generalization on larger mazes. y-axis uses probit transformation. (**right**) PI models are better able to make use of additional test-time computation. We trained models with varying number of training-time iterations, learning rate and weight norm application. Bit-wise accuracies are evaluated and averaged over different string-lengths.

Some architectures guarantee the path independence property (see Sec. 2). However, most common DEQ architectures—and the ones we use throughout this paper—have the expressive power to learn multiple fixed points per input. Since it is unclear whether architectures enforcing the contraction property lose expressiveness [Bai et al., 2021], we focus our investigation on unrestricted architectures.

PI networks represent a different model for computation than standard feed-forward networks: instead of learning an entire computational graph to map inputs to outputs, they only have to *learn where to stop*. We dedicate the rest of the paper on exploring the *Path Independence Hypothesis*—the idea that models which successfully fit the training distribution with a path independent function are better able to exploit more test-time iterations to achieve higher accuracy, compared to those which fit the training distribution with a path-dependent function.

Before establishing a connection between path independence and out-of-distribution generalization, we first describe two architectural components that are *necessary conditions* for achieving path independence. Afterward, we describe a metric to quantify how path-independent a trained network is.

### 4.1 Architectural Components Necessary for Path Independence

Past work has observed that *weight tying* and *input injection* are both crucial for upwards generalization [Bansal et al., 2022]. We observe that both architectural components are also necessary for a learned model to be PI.[6] Without weight tying, the network is constrained to have a fixed forward depth, so it is meaningless to talk about the limiting behavior in large depth. Input injection ensures that the equilibrium point depends on the input despite having an "infinite depth". Without input injection, a PI network would necessarily forget the input; hence, any model which successfully fits the training distribution must be path dependent.

Interestingly, both architectural motifs are also key components of deep equilibrium models [Bai et al., 2019]; in that work, the motivation was to enable efficient gradient estimation via the implicit function theorem (IFT) — a concept closely related to path independence, since the premise of the IFT gradient estimator is that only the final hidden state matters, not the path taken to get there. It is striking that two separate lines of work would converge on the same architectural motifs, one motivated by generalization and the other by a variant of path independence.

Reproducing the results of Bansal et al. [2022], in Fig. 2a we show upward generalization performance using both equilibrium models and progressive nets [Bansal et al., 2022] – and the lack thereof using non-input-injected networks. For the remainder of this paper, we focus on architectures with both input injection and weight tying.

---

[6]Our definition also admits non-input-injected models to be path independent if they're representing constant functions (i.e. input independent). We don't consider such cases in our analyses.

**Algorithm 1** Asymptotic Alignment Score

**Input:** A batched input $\begin{bmatrix} \boldsymbol{x}_1 \\ \boldsymbol{x}_2 \end{bmatrix}$, an operator $f_{\boldsymbol{w}}$

**Initialize:** $\begin{bmatrix} \boldsymbol{z}_1 \\ \boldsymbol{z}_2 \end{bmatrix} = \mathbf{0}$

**Define:** $h(\boldsymbol{y}_1, \boldsymbol{y}_2) = \dfrac{\boldsymbol{y}_1}{\|\boldsymbol{y}_1\|_2} \cdot \dfrac{\boldsymbol{y}_2}{\|\boldsymbol{y}_2\|_2}$

Compute $\begin{bmatrix} \boldsymbol{z}_1' \\ \boldsymbol{z}_2' \end{bmatrix} = \mathrm{FIX}_{f_{\boldsymbol{w}}}\left( \begin{bmatrix} \boldsymbol{x}_1 \\ \boldsymbol{x}_2 \end{bmatrix}, \begin{bmatrix} \boldsymbol{z}_1 \\ \boldsymbol{z}_2 \end{bmatrix} \right)$

# Interchange and reinitialize iterates

Compute $\begin{bmatrix} \boldsymbol{z}_1'' \\ \boldsymbol{z}_2'' \end{bmatrix} = \mathrm{FIX}_{f_{\boldsymbol{w}}}\left( \begin{bmatrix} \boldsymbol{x}_1 \\ \boldsymbol{x}_2 \end{bmatrix}, \begin{bmatrix} \boldsymbol{z}_2' \\ \boldsymbol{z}_1' \end{bmatrix} \right)$

**return** $\mathrm{average}(h(\boldsymbol{z}_1'', \boldsymbol{z}_1'), h(\boldsymbol{z}_2'', \boldsymbol{z}_2'))$

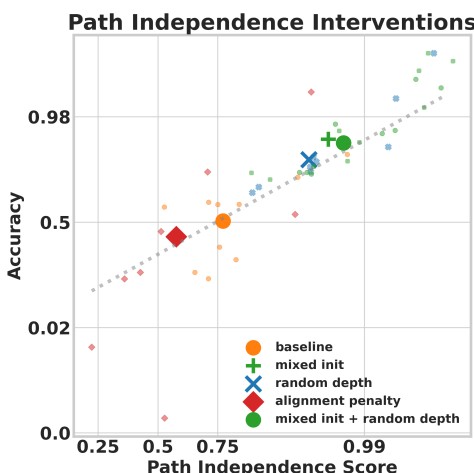

Figure 3: **(left) AA Score Algorithm:** We provide the algorithm for a simple illustrative case of two inputs. In practice, we consider larger batches. **(right) Promoting path independence improves generalization in the prefix sum task:** Interventions that are designed to promote path independence (initializing fixed points with random noise or running the fixed point solver with stochastic budget) improves generalization. Conversely, those that hurt path independence (penalty term that directly penalizes fixed point alignment) leads to poorer generalization.

## 4.2 Quantifying Path Independence

We propose a simple metric to quantify path independence based on the directional alignment of the fixed points computed with the same input, but different initializations. We name this metric the *Asymptotic Alignment (AA) score*. Pseudocode to compute the metric is given in Alg. 1. The AA score is the average cosine similarity between the fixed points obtained with the training time initialization (often simply the zero vector) and the fixed points obtained when one initializes the solver *using the fixed points computed on different inputs*. Higher AA scores (with 1 being the highest value) imply higher degrees of path independence. In Sec. 5, we show a strong correlation between path independence and accuracy using the AA score.

The AA score is cheap to compute, is a reliable indicator of path independence (see below), and is unitless, meaning that networks obtained from different training runs can be compared on equal footing. See the supplementary material for other metrics we've considered for quantifying path independence and why we found AA score to be preferable.

**Stress-testing the AA score**   To stress-test the extent to which the AA score really measures path independence, we search for *adversarial initializations* that are optimized to result in distinct fixed points, hence low AA values. (Unlike adversarial examples, this attack is not constrained to an $\varepsilon$-ball.) We use the L-BFGS [Liu and Nocedal, 1989] optimizer, and repeat the search multiple times starting from different fixed point initializations. We include pseudocode in the supplementary material.

Results of the adversarial stress test can be seen in Tab. 1. The results corroborate that the AA score is indeed a reliable measure of path independence; while it isn't possible to find adversarial initializations for high AA score networks (indicating high path independence), low AA score networks can easily be adversarially initialized to be steered away from the original fixed point estimate.

## 5   Path Independence Correlates with Upward Generalization

Is path independence (as measured by the AA score) a strong predictor of upwards generalization? We took the trained networks from Sec. 3, computed their average AA scores on in- and out-of-distribution splits and inspected whether the AA scores are correlated with upward generalization.

| Model | Task | AA ↑ | Accuracy (%) | Attacked AA ↑ | Attacked Acc. (%) |
|---|---|---|---|---|---|
| Non-PI network | Maze | 0.32 | 87.12 | 0.09 | 0 |
| PI network | Maze | 1.00 | 100 | 1.00 | 100 |
| Non-PI network | Prefix sum | 0.62 | 66.66 | 0.18 | 0 |
| PI network | Prefix sum | 0.99 | 100 | 0.99 | 100 |

Table 1: **Stress-testing the AA Scores:** AA scores for PI vs non-PI networks computed on $13 \times 13$ mazes and 64 bit prefix sum. Attacked AA refers to the cosine similarity between the fixed point from zero initialization and an adversarial initialization. Non-PI networks can be easily steered away from the initial fixed point estimate through adversarial initializations but it is difficult to do so for PI networks with high AA scores.

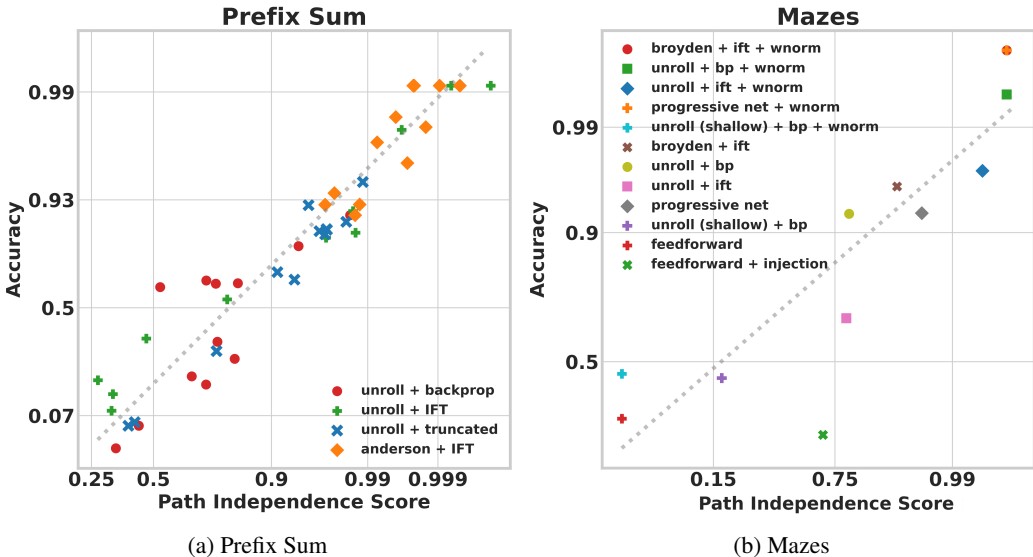

(a) Prefix Sum

(b) Mazes

Figure 4: High AA scores correlate with good upward generalization. For a given choice of an architecture and a task, the reported numbers are averaged over problem instances of different dimensions. We apply the probit transformation along both axes, following Miller et al. [2021]. Accuracies and AA scores are capped at 0.999 for compatibility with the probit transform.

On prefix sum experiments, we varied 1) network depth, 2) whether or not weight norm (wnorm) [Salimans and Kingma, 2016] was used or not,[7] 3) learning rate (one of [0.01, 0.001, 0.0001]), 4) forward solver (fixed point iterations or Anderson acceleration [Anderson, 1965], and 5) the gradient estimator (backprop or implicit gradients).[8] On the maze experiments, we varied 1) network depth, 2) use of weight norm, 3) forward solver (fixed point iterations or Broyden solver [Broyden, 1965]), and 5) the gradient estimator (backprop or implicit gradients).

Fig. 4 displays our findings. We evaluated performance on a mixture of in- and OOD validation data; results on individual data splits can be found in the supplementary material. The results show a strong correlation between AA score and accuracy when the inference depth is large enough. This shows that PI networks allow for scaling test-time compute to improve test-time accuracy (see also Fig. 2b). The in-distribution validation performance of non-PI networks degrades with deeper inference depths. Unsurprisingly, these networks generalize poorly on harder problem instances that require deeper inference depths (i.e. problem instances that provably require at least a given number of layers to handle). Further results on the BlurryMNIST, matrix inversion and edge copy tasks can be found in Supplementary Material H, I and J.

---

[7] Bai et al. [2019] report that weight norm helps stabilize the training of DEQ models.

[8] Note that the deep equilibrium model (DEQ) setup [Bai et al., 2019] correspond to using a root solver (such as Anderson) for the forward pass and implicit gradients for the backward pass.

# 6 Experimental Manipulations of Path Independence

The previous section demonstrates a strong correlation between path independence and the ability to exploit additional test-time iterations. Unfortunately, we can't make a causal claim based on these studies: the observed effect could have been due to an unobserved confounder. In this section, we intervene directly on path independence by imposing regularizers which directly encourage or penalize path independence. We find that interventions designed to *promote* path independence also improve generalization, while interventions designed to *reduce* path independence also hurt generalization.

## 6.1 Promoting Path Independence via Randomized Forward Passes

A straightforward way to encourage path independence is simply to initialize the hidden states with random noise during training. To this end, we experimented with initializing the hidden states with zeros on half of the examples in the batch, and with standard Gaussian noise on the rest of the examples. The reason to include the zero-initializations at training time is that we initialize from zeros at test time - not including this initialization during traning time causes a distribution shift.

Another way to promote path independence is simply running the forward solver with randomized compute budgets/depths during training time. While a path independent solution can be expected to be robust against this intervention, a path dependent one will fail.

We took the training configurations of the 12 prefix-sum networks described in Sec. 5 that use fixed point iterations in their forward pass, and backpropagation gradient in their backward pass, and retrained them separately with the aforementioned mixed initialization and randomized depth strategies without modifying any other experimental conditions. As can be seen in Fig. 3, the interventions lead to strong test-time path independent neural networks, while also reliably improving in- and out-of-distribution validation accuracy. We especially emphasize that shallow networks trained with mixed initialization actually remain far from having high AA scores using the training-time forward pass conditions due to lack of convergence. However, since the mixed initialization strategy results in path independent networks, scaling up test-time compute budget leads to high AA scores, and therefore high upwards generalization.

## 6.2 Penalizing Path Independence via the Fixed Point Alignment Penalty

Does an intervention that results in less path independence also result in poorer upwards generalization? Like in the mixed initialization experiment, we retrained the 12 unroll + backpropagation networks with an additional auxiliary loss term that penalizes the dot product between the fixed points computed from the same input, but different initializations sampled from standard Gaussian noise. Fig. 3 shows that this intervention succeeded in pushing the AA scores down, while also keeping the accuracy on the same trend line.

# 7 Disambiguating Convergence and Path Independence

Is convergence necessary for path independence? We answer this statement in the negative, and show that neither training-time convergence nor test-time convergence is required for path independence. Instead convergence to the same limiting behavior regardless of initialization is important.

**Training Time Convergence** We consider two implicitly trained equilibrium models trained on the mazes task—one trained with implicit gradients computed via implicit function theorem (IFT), and the other trained with an approximation of the (inverse) Jacobian, called phantom gradients [Geng et al., 2021b]. We report the values of residuals (*i.e.,* $\|f_{\boldsymbol{w}}(\boldsymbol{x}, \boldsymbol{z}) - \boldsymbol{z}\|_2$), AA scores and accuracies observed for in- and out-of-distribution data for mazes in Tab. 2. We observe that DEQs trained with phantom gradients have higher values of in-distribution residuals but are path independent, as indicated by their high AA scores, and show strong upward generalization as indicated by their good accuracy.

The mixed-initialization intervention described in Sec. 6.1 also leads to a separation between training-time convergence and path independence. We found that it is possible to train very shallow (*i.e.,* 5 layer) unrolled networks that, while being very far from converging during training and attaining poor in-distribution generalization, are able to converge and achieve perfect performance when run for many more iterations during test time. Details are provided in the supplementary material.

| Model | Residual ↓ | | AA score ↑ | | Accuracy (%) ↑ | |
|---|---|---|---|---|---|---|
| | In-dist | OOD | In-dist | OOD | In-dist | OOD |
| DEQ (phantom grad.) | 11.83 | 0.016 | 0.96 | 0.99 | 99.96 | 99.88 |
| DEQ (IFT) | 1.4 | 0.011 | 0.99 | 0.99 | 99.99 | 100 |

Table 2: Training-time convergence is not needed for path independence: models might show poor training-time convergence (as shown by high values of residuals) but still be path independent. Residual, AA score, and Accuracy for DEQ trained with IFT vs phantom gradients. In-distribution (In-dist) results were computed on $9 \times 9$ mazes, and OOD results were computed on $25 \times 25$ mazes.

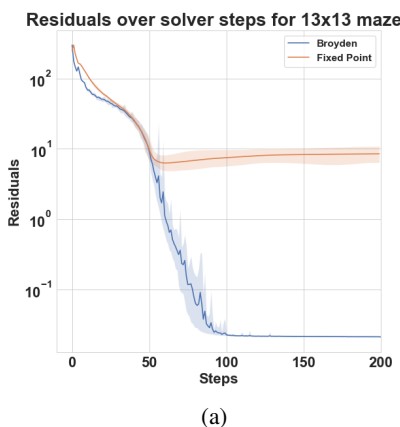
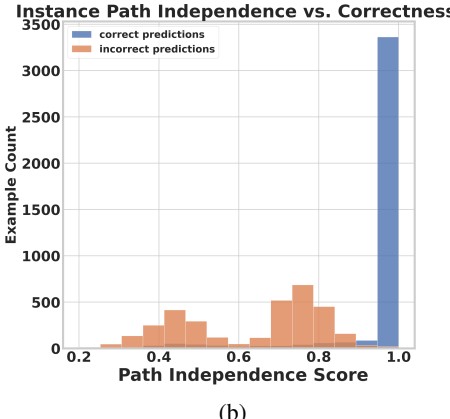

(a)  (b)

Figure 5: (**Left**) Different solvers display differing asymptotic behaviours but still achieve good upwards generalization. Here, the network has an adversarial AA score of 0.99, and achieves accuracy of 99.98% (fixed point iterations) and 99.97% (Broyden solver) respectively on the mazes task; (**Right**) Per-instance path independence is highly correlated with correctness of predictions for prefix sum task.

**Test Time Convergence** From Tab. 2, one might conclude that test time convergence is important for path independence. However, we show that this connection is not necessary, and convergence to the same fixed point is not a required condition for path independence. We study test time convergence properties of an unrolled weight-tied input-injected network trained with backpropagation under different solvers. This network is highly path independent using either the Broyden solver or fixed point iterations, as indicated by its high AA scores (0.99) on both in- and out-of-distribution data. We visualize the values of test-time residuals with fixed point iterations and Broyden's method in Fig. 5a. Both these solvers converge to different limit cycles but still show good upward generalization.

## 8 Path Independence on a Per-Example Level

The connection between path independence and prediction correctness also largely holds on a per-instance basis. Using the prefix-sum networks trained with the mixed-initialization strategy (the most performant group of networks in our intervention experiments), we plotted the distribution of per-instance fixed point alignment scores, colored by whether the prediction on that instance was correct or not in Figure 5b. This suggests that path independence can be used as a valuable sanity-check to determine whether a prediction is correct or not without the need for any label data, both in- and out-of-distribution. We provide a more in-depth per-instance analysis in the supplementary material.

## 9 Related Work

There is a long line of research on neural networks that can adapt their computational budget based on the complexity of the task they are learning to solve—akin to the intrinsic mechanism in humans to reason and solve problems. Schmidhuber [2012] introduced self-delimiting neural networks which

are a type of recurrent neural networks (RNNs) that adapt their compute based on the output of a special "halt" neuron. Adaptive computation time (ACT) [Graves, 2016a] also uses the output of a sigmoidal halting unit to determine the termination condition of an RNN, but it avoids long "thinking" time by explicitly penalizing it. Subsequent works have successfully applied variants of ACT in image classification and object detection [Figurnov et al., 2017], visual reasoning [Eyzaguirre and Soto, 2020], Transformers [Vaswani et al., 2017] for language modelling [Dehghani et al., 2019, Elbayad et al., 2020a, Liu et al., 2021b], and recognizing textual entailment [Neumann et al., 2016]. PonderNet [Banino et al., 2021] reformulates the halting policy of ACT as a probabilistic model, and adds a regularization term in the loss objective to encourage exploration. With these additions, PonderNet can extrapolate to more difficult examples on the parity task, first proposed by Graves [2016b]: in a vector with entries of 0, -1, and 1, output 1 for odd number of ones, and 0 otherwise. In this work, we do not optimize or penalize the network for the number of computational steps. Our main goal is to understand the underlying mechanism that results in scalable generalization of equilibrium models on harder problem instances. Our current work is closely related to previous work by Schwarzschild et al. [2021b] and [Bansal et al., 2022] that propose architectural choices and training mechanisms that enable weight tied networks to generalize on harder problem instances. We relate these papers' contributions to ours in Section 4.1.

Another family of models with the property of adaptive inference compute budget is early exit networks [Teerapittayanon et al., 2016, Laskaridis et al., 2021]. These networks have multiple additional "exit" prediction heads along their depth. At inference time, the result that satisfies an exit policy is selected as the prediction output. This approach of designing adaptive networks has been adapted both in natural language processing [Schwartz et al., 2020, Soldaini and Moschitti, 2020, Elbayad et al., 2020b, Zhou et al., 2020, Liu et al., 2020] and vision [Li et al., 2017, Wang et al., 2018, Xing et al., 2020, Kouris et al., 2021]. Most of these architectures have complex sub-modules that are trained in multiple stages, and require complex exit policies. In contrast, equilibrium models have a simple architecture, and can use root solvers to efficiently solve for the fixed point at inference.

More complex transformer-based language models like GPT-3 also struggle to generalize well on simple algorithmic tasks like addition [Brown et al., 2020]. Recent work by Nye et al. [2021] shows that transformers can be trained to perform well on algorithmic tasks and generalize on OOD data by emitting the intermediate steps of an algorithm to a buffer called "scratchpad". Using a scratchpad enables the model to revisit its errors and correct them.

Path independence is closely related to the concept of *global stability and global convergence* in control theory and optimization. This concept is somewhat overloaded, as it sometimes requires convergence to a single point [Slotine et al., 1991], and sometimes implies the system is convergent everywhere, even if to different points [Wang et al., 2003, Sriperumbudur and Lanckriet, 2009]. We thus choose the term *path independence* to refer specifically to the fact that the system will converge to the same limiting behavior (whatever that might be) regardless of the initial state of the system.

## 10    Conclusion

Being able to attain better levels of performance using a larger inference-time compute budget is a feat that eludes most standard deep learning architectures. This is especially relevant for tasks that require *upwards generalization*, *i.e.,* the ability to generalize from easy problem instances to hard ones. We show that equilibrium models are capable of displaying upwards generalization by exploiting scalable test-time compute. We link this to a phenomenon we call *path independence*: the tendency of an equilibrium network to converge to the same limiting behavior given an input, regardless of the initial conditions. We investigate this phenomenon through careful experiments and verify that path independent networks indeed generalize well on harder problem instances by exploiting more test time compute. Moreover, interventions on training conditions that promote path independence also improve upwards generalization, while those that penalize it hurt this capability. Our findings suggest that path independent equilibrium models are a promising direction towards building general purpose learning systems whose test-time performance improves with more compute.

## 11 Acknowledgements

AP is supported by a grant from the Bosch Center for Artificial Intelligence. JT acknowledges support from the Center on Long-Term Risk Fund. CA is supported by NSERC Canada Graduate School - Doctorate scholarship.

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
