# OpenReview forum: "Path Independent Equilibrium Models Can Better Exploit Test-Time Computation"
_NeurIPS.cc/2022/Conference — NeurIPS 2022 Accept_

### Official Review · Reviewer_nzeU · 2022-06-23

**Rating:** 4
**Confidence:** 3
**Soundness:** 2 fair
**Presentation:** 2 fair
**Contribution:** 3 good

**Summary:**

Deep equilibrium models (DEQs) are built from layers that solve fixed point equations. Two interesting features of DEQs are:
(1) Layers of DEQs in their most general form may have no solutions, a unique solution, many solutions, countably infinitely solutions, or uncountable infinitely many solutions, and (2) test-time computational budget may be chosen in a DEQ by trading off accuracy versus budget in the fixed point solver.

The authors empirically find that a certain measure of path independence (how independent the output of a DEQ layer is of initialisation) correlates well with the ability of the DEQ to utilise an increased test-time budget for benefit.

**Questions:**

- I do not understand FPA. Can you help me understand Algorithm 1? Let me summarise my understanding:
    1. Do one pass through f with an initial state z=0 for two different inputs, x1 and x2. This gives z1' and z2' respectively.
    2. Compute a second pass for (x1, z1'), giving z2''.
    3. Compute the cosine similarity between z2'' and z1'.
According to the notation used, no fixed points of any function are ever actually computed in Algorithm 1. As far as I can tell, z2' is never used, and z2 is only used to compute z2'? Perhaps my confusion is related to the comment "Interchange and reinitialize fixed points".
- After clarifying my understanding of FPA, can you explain why this is a useful measure?
- What is the motivation for seeking adversarial examples in the z-space? Does this represent a trimmed-down real life scenario? I can't imagine an adversary ever having access to the initial state of the fixed point solver.
- The first sentence in line 250. Why is this true? Won't the learned networks depend on the random initialisation? I don't think this statement is true in general. A counter-example is if the network is a contraction, then under mild conditions a reasonable fixed point solver (e.g. forward iteration) will converge to the unique equilibrium. So the network's output will not depend on the initialisation.
- Figure 7 caption and legend. What does "Fixed point solver" mean? Is this naive fixed point iteration? What does "residuals" mean? The equation on line 296 is not easy to read. previously f had two arguments, z and x. Here it only has one argument? Does this maybe mean ||f (x, z) - z||_2, which would indicate how close we are to a fixed point? Then is it true to say that if the residual is not close to zero, a fixed point has not been obtained? Then the first sentence of the caption in figure 7 is misleading, because in the case of naive fixed point iteration, a fixed point was not found?

**Limitations:**

The techniques discussed here are suitably general and do not have an application in mind.

**Strengths And Weaknesses:**

Strengths:
- It is unknown how the number of solutions that a DEQ layer admits interacts with the performance of a system. It is also unknown how to make best use of the fact that DEQs have a test-time computational budget. It is great to see research that attempts to show that path independent networks generalise well when they are able to exploit more computational budget at test time.
- I like the idea of the interventions performed in section 6. The execution is mostly great too, conditioned on some of my concerns detailed in the weaknesses and questions.

Weaknesses
- I did not enjoy the discussion about the fixed point iteration method at the start of section 2. The authors describe the fixed point iteration method, but not conditions under which it converges to a fixed point. More worrying is the implicit language which seems to say that it does converge to a fixed point (.e.g until it converges to a fixed point). A mention of contraction properties or Banach's fixed point theorem would be nice here, if fixed point iteration is to be discussed. This is later mentioned in footnote 2.
    - The weight tying view (mentioned around line 100 and elsewhere) is only valid if the DEQ can be interpreted as an infinite depth network. A sufficient condition is contraction/Banach's fixed point theorem, but in general, a fixed point condition cannot be interpreted as an infinite depth network with weight tying. Other works talk about other sufficient conditions.
- I do not understand the motivation for algorithm 1, nor do I see how it is a relevant metric. See Questions below.
- There is limited discussion and no mathematical theory about *why* the Path Independence Hypothesis might be true and how FPA relates to the hypothesis.
- I have several other concerns (which may be more to do with clarity than actual technical issues), detailed in questions below. If these are technical issues, they are in my opinion very significant ones. If they are clarity issues, then they are still significant and need to be addressed before this work is suitable for publication.

---

> ### Author Response · Authors · 2022-08-02
> **Response to Reviewer nzeU (2/2)**
>
> > _After clarifying my understanding of FPA, can you explain why this is a useful measure?_
>
> * We hope that the above clarifications have made it clearer why the FPA score measures path independence: If we initialize the forward pass with many fixed points obtained from different problem instances and **_still_ end up consistently finding the same fixed point** we did when we initialized with 0s, then we can conclude that the network is path independent on that particular problem instance.
> * Please see **Appendix A**, which explains our **careful selection process for quantifying path independence**.
> * The FPA score simply checks *directional similarity* and discards magnitude information. This makes the metric **dimensionless**, which is a requirement for it being comparable across different training runs (otherwise the metric would depend on network initialization, optimizer characteristics etc.)
> * As also explained in the general comments, FPA score is the most efficient metric that we’ve considered that can reliably tell whether a network is path independent.
>
> > _What is the motivation for seeking adversarial examples in the z-space? Does this represent a trimmed-down real life scenario? I can't imagine an adversary ever having access to the initial state of the fixed point solver._
>
> * The purpose of this experiment is **to check whether FPA is a reliable metric of path independence**. If the FPA score being high implies how path independent a network is, then we shouldn’t be able to find an adversarial initialization that makes the network path-dependent. We have updated Table 1 to include results of prefix sum. As we can infer from the table, it is indeed difficult to find an adversarial initialization on path independent networks but we can easily find adversarial initializations for non path independent networks.
>
> > _The first sentence in line 250. Why is this true? Won't the learned networks depend on the random initialisation? I don't think this statement is true in general. A counter-example is if the network is a contraction, then under mild conditions a reasonable fixed point solver (e.g. forward iteration) will converge to the unique equilibrium. So the network's output will not depend on the initialisation._
>
> * If we understand correctly, you’re referring to this sentence: “A straightforward way to reduce the dependency of learned networks on initialization is simply initializing the hidden states with random noise.”
> * To clarify: we meant **initializing the hidden states with random noise _during_ training** helps promote path independence. Unconstrained models can easily (and often do) learn functions that are neither contractions nor path-independent functions.  In this circumstance, random initialization completely messes up the forward pass, and the network become unusable. Training with this perturbation makes non-PI solutions non-viable, because they lead to poor losses. Similarly, if an unconstrained DEQ learns a contraction, then that’s good! It’s going to be robust against random initialization because it
>
> Does this answer your question? Please let us know if we’ve misunderstood your concern.
>
> > *Figure 7 caption and legend. What does "Fixed point solver" mean? Is this naive fixed point iteration? What does "residuals" mean? The equation on line 296 is not easy to read. previously f had two arguments, z and x. Here it only has one argument? Does this maybe mean $\||f (x, z) - z\||_2$, which would indicate how close we are to a fixed point? Then is it true to say that if the residual is not close to zero, a fixed point has not been obtained? Then the first sentence of the caption in figure 7 is misleading, because in the case of naive fixed point iteration, a fixed point was not found?*
>
> * Fixed point solver does mean naive fixed point iterations. We’ve fixed the caption. We have also updated the equation in Line 296. Indeed, the equation should be $\||f (x, z) - z\||_2$.
> * As you’ve guessed, “residual” stands for the Euclidean distance between subsequent solver iterates.
> * It is true that in the case of naive fixed point iterations, a fixed point has not been found and instead we converge to limit cycles. We have added additional analysis on this in Section G of the appendix. We provide per-instance analysis for the values of residuals and L2 norm between the fixed points of Broyden solver and naive fixed point iterations. Our plots clearly indicate that there are problem instances where **naive fixed point iterations converge to a limit cycle** but output correct predictions which validates our observation that test time convergence is indeed not necessary. On the same problem instances, Broyden clearly displays a different asymptotic convergence behavior.
>
>
> [1] Revay, Max, Ruigang Wang, and Ian R. Manchester. "Lipschitz bounded equilibrium networks." arXiv preprint arXiv:2010.01732 (2020).

---

> > ### Comment · Reviewer_nzeU · 2022-08-08
> > **thanks for your response**
> >
> > I'm not satisfied with the explanation of FPA being a useful measure. Numbers referring to your appendix A:
> > 1. There are an uncountable number of dimensionless measures. Viewed as a kernel, the normalised inner product (cosine angle) is just a normalised linear kernel. What about using a general non-linear kernel, such as RBF? These would also be "dimensionless". Also, I don't necessarily see how being dimensionless means that metrics computed from different training runs should be directly comparable with each other (and presumable a dimensioned quantity would not be comparable?)
> > 3. Efficiency. See, for example, RBF in point 1.

---

> > > ### Author Response · Authors · 2022-08-08
> > > **Thank you for your follow-up**
> > >
> > > Thank you for your follow-up.
> > >
> > > **We’ve updated the paper to integrate our answers to your questions.**
> > >
> > > **Other useful metrics:** Our commitment to the FPA score only stems from its appealing properties (like unitlessness), empirical success (in quantifying how similarly the limiting behaviours are on different initializations, locally and globally) and efficiency. Other metrics that satisfy these criteria can also be used to quantify path independence.
> > >
> > > In other words, our results should not depend on the specific implementation details of path independence metrics (as long as they satisfy the criteria we set out in Appendix A). We confirmed this via an experiment described below.
> > >
> > > **Other sensible metrics behave similarly to FPA scores:** Following your suggestion, we tried using other RBF flavoured kernels to quantify path independence. Concretely, we tried the Gaussian, Laplacian and multiquadratic kernels. To make things unitless, we first normalized the fixed points, then computed the pairwise similarities (please see below for why unitlessness is important). **We’ve added this analysis Appendix A - please see the newly added plot.**
> > >
> > > The takeaways from the experiments remain identical: **path independence is strongly correlated with generalization, regardless of the specific details of how path independence is quantified** (as long as it satisfies the criteria we set in Appendix A).
> > >
> > >
> > > **Importance of unitlessness:** Being unitless is **necessary** for a metric to be comparable across training runs — though obviously not sufficient. Consider two equilibrium models M1 and M2, where the fixed points computed by M2 have the same direction as those computed by M1, but with twice the Euclidean norm. The behaviour of this M2 is qualitatively the same as that of M1, but any metric that depends on the Euclidean metric (or Jacobian L2 norm, or other otherwise sensible non-unitless metric) would report this network to be less path independent. Note that this is not a purely theorical consideration: simply using LayerNorm, adding L2 regularization, or penalizing the magnitude of fixed points to encourage convergence will directly impact the scale of the fixed points (things that practitioners often use), thereby rendering non-unitless metrics unreliable. **We added this discussion in the paper.**
> > >
> > > **Efficiency:** Cosine similarity and the other alternative kernels described above are all significantly more efficient compared to the alternatives we considered in the Appendix.
> > >
> > > We hope that these have addressed your major concerns. Please let us know if you have any remaining concerns, and consider if these improvements in the paper amount to an increase in your score.

---

> ### Author Response · Authors · 2022-08-02
> **Response to Reviewer nzeU (1/2)**
>
> Thank you for your careful review and attention to detail. We believe we have solid answers to the points you’ve brought up, and have already improved the paper in light of your critique. Please see below (as well as the improvements in the paper), and consider increasing your score if they sufficiently address your concerns!
>
> **Summary**
> * **Improved discussion of convergence**: New paragraph in Section 2 that states convergence conditions for fixed point iterations and DEQs
> * **Clarification of FPA calculations**: Updated Algorithm Box 1 to clarify the procedure for calculation of Fixed point alignment (FPA) score. FPA score is useful because it is reliable, efficient and unitless (see our response below)
> * **Extended per-instance analysis of test time convergence**: New experiments on per-instance analysis of test time convergence in Section G of appendix
>
>
> **General Response**
>
> Most of the concerns below actually seem to be due to clarity issues that are already fixed in the updated version of the paper. We’ve also improved the writing and exposition substantially. Please let us know what you think!
>
> > _I did not enjoy the discussion about the fixed point iteration method at the start of section 2. The authors describe the fixed point iteration method, but not conditions under which it converges to a fixed point. More worrying is the implicit language which seems to say that it does converge to a fixed point (.e.g until it converges to a fixed point). A mention of contraction properties or Banach's fixed point theorem would be nice here, if fixed point iteration is to be discussed. This is later mentioned in footnote 2.
> The weight tying view (mentioned around line 100 and elsewhere) is only valid if the DEQ can be interpreted as an infinite depth network. A sufficient condition is contraction/Banach's fixed point theorem, but in general, a fixed point condition cannot be interpreted as an infinite depth network with weight tying. Other works talk about other sufficient conditions._
>
> We’ve **improved the way we discuss convergence**. Here are the main updates:
> * We added a paragraph discussing convergence conditions in Section 2. Earlier, this was casually discussed in Section 4.
> * We mention some sufficient conditions for convergence, such as global contractivity (which can be achieved by imposing a Lipschitz constraint on the recurrent cell, as Revay et. al. [1] does) and the MONDEQ approach.
> * We clarify that the implicit formulation of DEQs are more general than the “infinite depth” formulation, as it admits unstable fixed points that could be found by certain solvers. (Alternatively, one can view DEQs as infinitely deep “solver-wrapped” (i.e. with Broyden or Anderson) depthwise recurrent networks.)
> * Lastly, we clarify that divergence is also an option: the chosen black-box solver (or simply fixed point iterations) could diverge, rendering the forward pass unusable.
>
> > _I have several other concerns (which may be more to do with clarity than actual technical issues), detailed in questions below. If these are technical issues, they are in my opinion very significant ones. If they are clarity issues, then they are still significant and need to be addressed before this work is suitable for publication._
>
> We’ve addressed all of the concerns you’ve raised below, which we believe were due to clarify issues. Please follow-up if you find any response unsatisfying.
>
> **Questions:**
>
> > _I do not understand FPA. Can you help me understand Algorithm 1? Let me summarise my understanding:_
> > _* Do one pass through f with an initial state z=0 for two different inputs, x1 and x2. This gives z1' and z2' respectively._
> > _* Compute a second pass for (x1, z1'), giving z2''._
> > _* Compute the cosine similarity between z2'' and z1'. _
> > _According to the notation used, no fixed points of any function are ever actually computed in Algorithm 1. As far as I can tell, z2' is never used, and z2 is only used to compute z2'? Perhaps my confusion is related to the comment "Interchange and reinitialize fixed points"._
>
> We’ve **updated the Algorithm Box 1** to make the procedure as clear as possible. The most crucial clarifications are: 1) What appears as $f_w()$ in the Algo box should have been $\mathrm{FIX()}$. All the zs stand for fixed points. 2) The “interchange and reinitialize” simply refers to recomputing the fixed points $[x1, x2]$ using the swapped $[z2’, z1’]$ as initializations.

---

> > ### Comment · Reviewer_nzeU · 2022-08-08
> > **Thanks for your resonse**
> >
> > Thanks for  your updates. I still have some concerns.
> >
> > Regarding updates to fixed point discussion:
> > In line 81, before you mention any considerations for fixed point iteration to be principled, you mention recursively applying a function until the procedure converges. The uninformed reader will not properly appreciate at this point that this procedure is not guaranteed to converge.
> > The details in the convergence paragraph around line 110 should be introduced before statements around line 81.
> >
> > FPA:
> > Thanks for fixing the issues with FPA, which seemed to have also thrown other reviewers.

---

> > > ### Author Response · Authors · 2022-08-08
> > > **Thank you for your follow-up**
> > >
> > > Thank you for your follow-up comments!
> > >
> > > **We've updated the writeup to reflect your recommendation.**
> > > * We made it clear that convergence only happens under certain condition early on (around line 81).
> > > * We explicitly stated that divergence is an option as well.
> > > * For the more general discussion of convergence, we referred the readers to two paragraph below.
> > >
> > > Please let us know if the clarifications above resolve your concern related to the discussion of convergence, or if you have any remaining concerns!

---

### Official Review · Reviewer_hf7z · 2022-07-01

**Rating:** 6
**Confidence:** 3
**Soundness:** 3 good
**Presentation:** 2 fair
**Contribution:** 3 good

**Summary:**

This paper studies the path independence of equilibrium models, which refers to the fact that a system will converge to the same limiting behavior regardless of the initial state. It hypothesizes that the path independence strongly correlates with generalization performance on harder examples, and then it experimentally verifies the hypothesizes. This paper also introduces a metric for measuring the path independence of a learned model, called FPA score. It also experimentally verifies that FPA strongly correlates with model accuracy on out-of-distribution samples.

**Questions:**

1. In Sec 4.1, the paper says, weight tying and input injection are both necessary for a learned model to be PI. However, according to the definition of path independence in L36-38:
We thus choose the term path independence to refer specifically to the fact that the system will converge to the same limiting behavior (\textbf{whatever that might be}) regardless of the initial state of the system.

PI doesn't need the limiting behavior to be meaningful. Without input injection, the limiting output would be a representation independent of the input, but it still satisfies the definition of path independence. As a result, I don't agree with the statement.

2. Wrong references: (L93-95) Recent works [Geng et al., 2021a,b, Fung et al., 2021] have shown that the inverse-Jacobian term in Eq. (1) can be replaced with an identity matrix without affecting the final performance of DEQs.

What the author describes is the work of Fung et al., not of Geng et al., 2021a,b. Geng et al., 2021b uses unrolling-based estimates of the inverse-Jacobian as a replacement, not an identity matrix. And I am a little confused why the author cites Geng et al., 2021a here.

**Limitations:**

No.

The author writes: (L727-737 in the Appendix) While the necessity of additional test-time compute is obvious for prefix sum and mazes, it’s less clear for perception-like tasks like image classification.
In my opinion, the stopping criterion used by DEQ itself is a signal to measure the necessity. To be specific, DEQ sets a stopping tolerance, once the error/relative error goes below it, the solver stops iterating. Maybe the iteration step taken by the solver can be used to decide if additional test-time computation is needed.

**Strengths And Weaknesses:**

Strengths:
1. Adaptive computational budget is an interesting problem and worth studying.

Weaknesses:
My main concerns about this paper concentrate on the paper writing. I have difficulty understating the following points:

1. Algorithm 1 doesn't coincide with the language description. There is neither a fixed point equation nor a root-solving process.
Also, since FPA seems to be a random metric, can the author show the variance introduced by different inputs?

2. In Figure 5 (a), where is the comparison between different solvers? Does unroll mean fixed point iteration?

3. L143-144, However, our definition allows for other behaviors such as limit cycles (see Section 7).
However, when I turn to Sec. 7, I don't find anything related to other behaviors except convergence to a fixed point.

4. I cannot understand what this means:
(L34-36) However, the term “global stability” is slightly overloaded, as it sometimes refers to the actual convergence, sometimes requires convergence to a single point, and sometimes just implies the system is convergent everywhere, even if to different points.

Could you please explain what the three cases refer to?

5. I cannot understand the caption of Figure 2:
Blue and grey dotted lines indicate in-distribution data and harder problem instances, respectively.
Then why do they fill 0 to 1 ?

---

> ### Author Response · Authors · 2022-08-02
> **Response to Reviewer hf7z (2/2)**
>
> > _L143-144, 'However, our definition allows for other behaviors such as limit cycles (see Section 7)'. However, when I turn to Sec. 7, I don't find anything related to other behaviors except convergence to a fixed point._
>
> * Figure 7a actually is a demonstration of **different solvers entering two different limit cycles**: the absolute residuals depicted in the figure are nonzero (this value would have been close to 0 upon convergence). **We ran additional per-instance analyses that without a doubt demonstrate limit cycle behaviour**. We find that there are problem instances where naive fixed point iteration clearly converges to a limit cycle but can solve the tasks correctly and also display upward generalization. In addition, on these particular problem instances, Broyden solver clearly displays a different asymptotic behavior but still solves the tasks correctly. Please see the new plots in Appendix D.
>
>
> > _I cannot understand what this means: (L34-36) However, the term “global stability” is slightly overloaded, as it sometimes refers to the actual convergence, sometimes requires convergence to a single point, and sometimes just implies the system is convergent everywhere, even if to different points. Could you please explain what the three cases refer to?_
>
> * **We’ve added references to the paper that provide additional clarification.**
> * Applied Linear Control by Slotine and Li (1990) discuss global stability in the context of 1) convergence to fixed points (chapter 3.2) convergence to limit cycles (chapter 3.4) and convergence to invariant sets (chapter 4 in general).
> * Quoting Sriperumbudur et. al. [1] “Note that the word “global convergence” is a misnomer. We’ll clarify it below and introduce some notation and terminology. [...] Note that depending on the objective and constraints, the minimized of the CCCP algorithm in (2) need not be unique. [...] Hence, the notion of point-to-set maps appear naturally in such iterate algorithms.“ The authors raise a similar concern to ours and specify that global stability refers to convergence to a set of points. This is exactly the type of behaviour we’d like NOT included in the definition of path independence.
>
> [1] Sriperumbudur, Bharath K., and Gert RG Lanckriet. "On the Convergence of the Concave-Convex Procedure." Nips. Vol. 9. 2009.
>
> > _I cannot understand the caption of Figure 2: Blue and grey dotted lines indicate in-distribution data and harder problem instances, respectively. Then why do they fill 0 to 1 ?_
>
> Blue line indicates that the network was trained on 9X9 mazes and then tested on larger mazes as indicated by grey lines. The lines fill 0 to 1 purely for aesthetic purposes. **We have updated the caption to reflect this.**
>
> > _In Sec 4.1, the paper says, weight tying and input injection are both necessary for a learned model to be PI. However, according to the definition of path independence in L36-38: We thus choose the term path independence to refer specifically to the fact that the system will converge to the same limiting behavior (\textbf{whatever that might be}) regardless of the initial state of the system. PI doesn't need the limiting behavior to be meaningful. Without input injection, the limiting output would be a representation independent of the input, but it still satisfies the definition of path independence. As a result, I don't agree with the statement._
>
> Thank you for bringing this up. We’re aware of this subtlety – we didn’t expand on it in the paper, as we believed it’d be **assumed that the networks we’re dealing with are computing input-dependent functions**. In order to do justice to this edge case, however, we added a footnote outlining this subtlety.
>
> > _Wrong references: (L93-95) Recent works [Geng et al., 2021a,b, Fung et al., 2021] have shown that the inverse-Jacobian term in Eq. (1) can be replaced with an identity matrix without affecting the final performance of DEQs. What the author describes is the work of Fung et al., not of Geng et al., 2021a,b. Geng et al., 2021b uses unrolling-based estimates of the inverse-Jacobian as a replacement, not an identity matrix. And I am a little confused why the author cites Geng et al., 2021a here._
>
> Thank you for pointing this out - **we’ve improved the text and citations**. We have updated the text to indicate that the backward pass can be Jacobian free [Fung et. al. 2021, Geng et. al. 2021a] or use an approximation to inverse-Jacobian [Geng et. al. 2021b]. We would also like to clarify that we cite Geng et al., 2021a because they approximate the inverse-Jacobian term with an identity. (Eq 13 in the version of paper on OpenReview)

---

> > ### Comment · Reviewer_hf7z · 2022-08-07
> > **Thanks for the authors' rebuttal**
> >
> > Thanks for the authors' rebuttal. I have read the authors' response and the revised version, and my major concerns have been addressed, so I will increase my score.

---

> ### Author Response · Authors · 2022-08-02
> **Response to Reviewer hf7z (1/2)**
>
> Thank you for your constructive criticism and insightful questions. We believe we have solid answers to almost all of your concerns - please see below and and consider if our response amounts to an increase in your score.
>
> **Summary**:
> * **Clarification of FPA calculations**: Updated Algorithm Box 1 to clarify the procedure for calculation of Fixed point alignment (FPA) score.
> * **Extended per-instance analysis of test-time convergence**: New experiments on per-instance analysis of test time convergence in Section G of appendix that clearly shows limit cycles.
> * **Additional experiments with multiple solvers**: Updated Figure 5(a) with results for multiple solvers.
> * **Explanation regarding global stability**: Updated list of references and citations to explain the ambiguity around the term ‘global stability’: Applied Linear Control by Slotine and Li, Sriperumbudur et. al. [1]
>
> **General Response**
>
> > _Algorithm 1 doesn't coincide with the language description. There is neither a fixed point equation nor a root-solving process._
>
> Thank you for pointing this out. We’ve **updated the Algorithm box 1**. We’ve replaced $f_w$ with a $\mathrm{FIX(x, z)}$ operation, which refers to the fixed point finding process. We’ve also formally defined this operator in the Background section.
>
> > _Also, since FPA seems to be a random metric, can the author show the variance introduced by different inputs?_
>
> **We added a discussion of our careful selection process and alternative metrics.** We actually did experiment with different metrics and methodologies to quantify path independence.  In summary, **FPA score is reliable, efficient and unitless** (allowing comparisons between trained networks) - qualities other methods we considered lack. Please see Appendix A where we discuss this in depth. Here’s a quick summary of other options we considered:
> * We tried computing the **Jacobian norm** between the output of the equilibrium model and its initialization. We found that this quantity vanishes on PI networks. This method is, unfortunately 1) extremely computationally intensive, 2) has “units”, meaning that values between different networks cannot be directly compared.
> * We also **tried something akin to your suggestion of using variance to quantify how PI a network is**: we computed the average squared Euclidean distance between the fixed points computed using differing initializations. Since this quantity is unitful, it doesn’t allow for cross-comparisons between different networks (the use of LayerNorm partially mitigates this, but not completely).
>
>   * Since the fixed points have very high dimensionality, we’re guessing that by variance, you mean the trace of the covariance matrix. This is equivalent to the squared Euclidean distance metric described above.
>
> > _In Figure 5 (a), where is the comparison between different solvers? Does unroll mean fixed point iteration?_
>
> * Yes, unroll means fixed point iteration.
> * We’ve **updated the figure with results for Broyden solver** too. The new results are consistent with our earlier findings.

---

### Official Review · Reviewer_VRFp · 2022-07-09

**Rating:** 6
**Confidence:** 2
**Soundness:** 3 good
**Presentation:** 4 excellent
**Contribution:** 2 fair

**Summary:**

This submission hypothesizes that there is a strong correlation between generalization and the so-called path independence (PI)--essentially positing that the state given the input and weights should be ``stable''. The authors make three major contributions:

1. They show that PI networks (typically deep equilibrium models) exhibit better upwards generalization when compared to non-PI models.
2. They propose a simple-to-compute metric, called the Fixed Point Alignment (FPA) score, to measure the level of path independence. They further demonstrate that FPA is highly correlated with upwards generalization.
3. To further test the validity of the path independence hypothesis, the authors design an experiment to adversarially intervene the training of PI models, and show that FPA is a robust metric for path independence.

**Questions:**

1. As alluded to above, do these PI models generalize as well if they are challenged with a task with, say, only 90% baseline accuracy?

2. Although the authors argue that there is no need for convergence in Section 7, I find it contradictory with the proposed FPA, which is clearly a measure of convergence. If, say, the model converges to a limit cycle, how does FPA detect it?

**Limitations:**

A major limitation of the submission is that the authors focus exclusively on the PI hypothesis for deep equilibrium models. However, PI is a quite general statement and, if true, should hold for non-deep equilibrium models as well. Figuring out a framework to incorporate the non-deep equilibrium models thus presents an interesting direction.

**Strengths And Weaknesses:**

In my opinion, this is a well-written paper with clear contributions. The authors have clearly laid out the thesis and nicely demonstrated the empirical evidence. The proposed FPA is intuitive and easy to compute, which might present an interesting metric to practitioners.

On the other hand, as a purely empirical work, I feel like we need more extensive experiments to conclude the validity of the proposals. The authors chose a benchmark dataset by Schwarzschild et al, which seems relatively easy and hence does not suffice to substantiate the PI hypothesis. In particular, it seems easy to get to essentially perfect accuracy on the proposed tasks, which poses the question that if these PI models would generalize as well if they are challenged with a task with, say, only 90% baseline accuracy?

---

> ### Author Response · Authors · 2022-08-02
> **Response to Reviewer VRFp (2/2)**
>
> > _On the other hand, as a purely empirical work, I feel like we need more extensive experiments to conclude the validity of the proposals. The authors chose a benchmark dataset by Schwarzschild et al, which seems relatively easy and hence does not suffice to substantiate the PI hypothesis. In particular, it seems easy to get to essentially perfect accuracy on the proposed tasks, which poses the question that if these PI models would generalize as well if they are challenged with a task with, say, only 90% baseline accuracy?_
>
> * **Why the current task?** The prefix sum and maze tasks have the appealing property that example difficulty can be dialed up indefinitely, forcing networks to (provably) spend more time on more difficult examples to solve them. This is why we mostly base our conclusions on these tasks.
> * **New results on BlurryMNIST**: Following your recommendation, we also tested our hypothesis on an image task called “Blurry MNIST” (train on clean images, test on images corrupted with differing levels of blur). We still see the same effect: higher levels of path independence indicate better generalization.
> * We’d also like to emphasize that the **Maze task proposed by Schwarzschild is not an easy task**! Learning to output optimal solutions to 59x59 mazes by only learning from 9x9 is an impressive capability that has only recently been achieved with equilibrium models and the deep thinking networks (which are also path independent) proposed by Bansal et. al. 2022 [1]
>
> > _A major limitation of the submission is that the authors focus exclusively on the PI hypothesis for deep equilibrium models. However, PI is a quite general statement and, if true, should hold for non-deep equilibrium models as well. Figuring out a framework to incorporate the non-deep equilibrium models thus presents an interesting direction._
>
> * Unfortunately, there’s a bit of a concept overload when it comes to discussing equilibrium models in the literature that we propagate in this submission. We’ll make clarifying modifications to the terminology.
> * Bai et. al. define “deep equilibrium models” (DEQ) to refer to the fully implicit formulation (i.e. black-box root finder for forward pass, implicit gradients for backwards pass). **We explore models beyond the implicit DEQ formulation**, as we also report results obtained using unrolling for forward pass and direct backprop gradients (and their combinations)
> * As we argue in section 4.1 (Architectural Components Necessary for Path Independence), **weight sharing and input injection are necessary for path independence**. As it happens, these are also exactly the requirements to train equilibrium models.
> * (Also note that several architectures proposed in the past few years that mimic optimization processes (like Differentiable Convex Optimization Layers [Agrawal et. al. (2019)][2] can also be viewed as equilibrium models, where the recurrent cell has a much more rigid structure.)
>
>
> [1] Bansal, Arpit, et al. "End-to-end Algorithm Synthesis with Recurrent Networks: Logical Extrapolation Without Overthinking." arXiv preprint arXiv:2202.05826 (2022).
>
> [2] Agrawal, Akshay, et al. "Differentiable convex optimization layers." Advances in neural information processing systems 32 (2019).

---

> > ### Comment · Reviewer_VRFp · 2022-08-08
> > **Thank you for your response**
> >
> > I thank the authors for their comprehensive response.
> >
> > - While I understand the authors' explanation regarding the usefulness of FPA towards limited cycles, it still seems to me that FPA is conceptually problematic. The empirical evidence provided by the authors is nice but not conclusive.
> >
> > - I understand that the maze task by Schwarzschild et al. is not an "easy" easy task, but the bottom line is that all tasks considered in the previous version have 99.95+% in-dist accuracy, which calls for the question: Does FPA really align with "general" generalization (meaning training error $\simeq$ test error always) or it merely detects "benign" memorization (training error $\simeq$ test error **only when** training error $\simeq 0)$? The newly proposed Blurry MNIST task doesn't seem to address this question.
> >
> > Notwithstanding these criticisms, I still like the paper and would recommend accept.

---

> > > ### Author Response · Authors · 2022-08-08
> > > **Thank you for your follow-up**
> > >
> > > Thank you for your follow-up and support.
> > >
> > > **We have further updated the paper based on your recent feedback.** Please take a look!
> > >
> > > **In-distribution error on BlurryMNIST is nonzero:**
> > > The in-distribution validation error actually varies between 1 and 5 percent between different models (i.e. not trivially close to $0$). Hence, we believe our results on BlurryMNIST is closer to measuring "general" generalization, albeit in a simplistic setup. We’ve added this clarification in the Supplementary Material.
> > >
> > > **Concerns related to FPA scores:**
> > > * We ran further evaluations where we replaced the cosine similarity operation with other kernels (i.e. Gaussian, Laplacian and inverse multiquadratic). **The results indicate that our takeaways are insensitive to the precise implementation details of the metric used to quantify path independence**, as long as the metric satisfies certain criteria we discuss in the paper. Our analysis is added in Supplementary Material A.
> > >
> > > **Further results on Matrix Inverse:** We also trained equilibrium models on the Matrix Inversion task introduced by Du et. al. [1] in their recently released (after NeurIPS deadline) paper. This task is concerned with learning to invert 20x20 matrices. Success is defined by how well the trained model works on matrices with worse condition numbers than those observed during training. Note that **this task is qualitatively different from all the others we considered before**. We’ve added the results in Supplementary Material I. Our takeaways are:
> > > * **Equilibrium models beat the baselines:** We beat all the baselines discussed in [1] and match the performance of the method proposed in [1].
> > > * **Poor PI => Poor Generalization:** We see the same trend: lower degrees of path independence ends up with poorer generalization.
> > >
> > > Please let us know if you have any concerns we could further address.
> > >
> > > [1] Du, Yilun, et al. "Learning Iterative Reasoning through Energy Minimization." International Conference on Machine Learning. PMLR, 2022.

---

> ### Author Response · Authors · 2022-08-02
> **Response to Reviewer VRFp (1/2)**
>
> Thank you for your encouraging review and insightful questions! We’ve significantly improved the paper, and have addressed your comments below.  Please take a look and see if they amount to an increase in your score!
>
> **Summary**
> * **Clarification of FPA calculations**: Updated Algorithm Box 1 to clarify the procedure for calculation of Fixed point alignment (FPA) score.
> * **Extended per-instance analysis of test-time convergence**: New experiments on per-instance analysis of test time convergence in Section G of appendix that clearly shows limit cycles.
> * **New results on image tasks**: We ran experiments one an image-based tasks (BlurryMNIST) and also observed a correlation between path independence and accuracy.
>
> **General Response**
>
> > _Although the authors argue that there is no need for convergence in Section 7, I find it contradictory with the proposed FPA, which is clearly a measure of convergence. If, say, the model converges to a limit cycle, how does FPA detect it?_
>
> * **FPA doesn’t consider magnitude convergence**: FPA measures directional similarity as opposed to explicit convergence, which ignores magnitude convergence. This makes FPA scores unitless, allowing comparisons between different networks meaningful.
> * **Movements within limit cycles are constrained in their radii**: We observe that limit cycles are often **localized**. For instance, the absolute residuals between the points in the limit cycle we discuss in Figure 7 are a small percentage of the Euclidean norms of the points: 0.9% for 9x9 mazes, 0.49% for 13x13 mazes and 1.5% for 25x25 mazes. Points in limit cycles don’t jump around to different directions, which preserves high FPA scores.  Please see our per-instance analysis in Appendix G.

---

### Official Review · Reviewer_4wU9 · 2022-07-11

**Rating:** 4
**Confidence:** 3
**Soundness:** 2 fair
**Presentation:** 3 good
**Contribution:** 2 fair

**Summary:**

This paper proposes the concept of 'path independence'. It proposes FPA score to measure the 'path independence' of a network, and shows that the FPA score is correlated with the upward generalization ability of equilibrium networks. The claims are supported by experimental results.

**Questions:**

1. Please correct me if there is any misunderstanding, but I do not understand the meaning of involving $x_2,z_2,z_2'$ in Algorithm 1. It seems to me that the algorithm only needs $z_1'$ in the followed steps, and the computation of $z_1'$ is not affected by $x_2,z_2$.
2. To show that FPA score is a valid measure for path independence, the authors compare the FPA and attacked FPA in table 1. However, this result is not promising enough. First, this table only shows two cases (PI and non-PI), but FPA score is a continuous measure. Two instances cannot show whether the FPA score is a valid measure as a continuous number. Second, for the non-PI network, the gap between the FPA score and attacked FPA score is large. Such a large gap makes it even harder to see how good FPA is as a measure for path independence. Especially, for non-PI networks, what does it mean if one has a higher FPA score than another? It is important to explain this.
3. In Figure 7 (b), the figure also indicates that when FPA score is less than 1, the correctness of the non-PI network is NOT always positively correlated with the FPA score.
4. Figure 7 (a) shows the residuals given by the two different solvers are different. However, this number is not rigorous enough to support the show that different solvers converge to different fixed points. It is better to directly compute the RELATIVE difference between the solutions given by two different solvers. Besides, to claim that this network is path independent, it will be more promising to show that its ATTACKED FPA score is close to 1, instead of the non-attached FPA score.

**Limitations:**



**Strengths And Weaknesses:**

Strength:

1. The paper is easy to follow and clear.
2. The phenomenon is interesting. The problem setting is potentially useful in practice.

Weakness:

1. All the claims are only supported by experiments, without any theoretical justifications.
2. The experiments are not rigorous enough.

---

> ### Author Response · Authors · 2022-08-02
> **Response to Reviewer 4wU9 (2/2)**
>
> **Questions:**
> > _Question regarding meaning of x2, z2, z2’_
>
> Note that **x1 and x2 are two different problem instances/inputs**. The idea here is to initialize the forward pass on x2 with the fixed point obtained using x1. In other words, we’re first computing fixed points using x1 and x2, then swapping them during another forward pass.** If the networks still find the same fixed point despite the swap, then we conclude it’s path independent**. In practice, we simultaneously swap multiple fixed points to get better statistical power.
>
> >  _”To show that FPA score is a valid measure as a continuous number.”_
>
> **More attacked FPA results and a graph added**: We’ve augmented our table with more results using models trained also on the prefix sum task. The outcome of these experiments is unchanged.
>
> * _”Second, for the non-PI network, the gap between the FPA score and attacked FPA score is large. Such a large gap makes it even harder to see how good FPA is as a measure for path independence. Especially, for non-PI networks, what does it mean if one has a higher FPA score than another? It is important to explain this.”_
>
> * **A gap expected when network is non-PI**: We indeed expect a gap between the FPA scores of non-attacked and attacked non-PI networks. Note that the attack is directly trying to bring down the FPA score. By analogy with adversarial examples, we'd expect adversarial attacks to be more successful than naive forms of perturbations of the same magnitude, even on adversarially robust networks.
> * **No gap when network is PI**: Even very strong attacks like L-BFGS isn’t able to bring down the FPA score of PI networks, confirming high FPA scores indicate path-independence (we believe this is quite surprising!).
> * **Meaning of high vs. low FPA scores**:  While the majority of the FPA scores we report in the paper are population averages, we depict in Section 8 that the relation between correctness and path-independence holds at a per-instance level. Lower levels of FPA scores indicate that a network will be non-PI on more problem instances, likely achieving poorer generalization.
> * We have updated Table 1 to include results of adversarial stress testing for Prefix sum task. Our observations for prefix sum are consistent with our previous observations with mazes: non-PI networks can be easily attacked through adversarial initialization but finding adversarial initialization for PI networks is difficult.
>
> >  *“In Figure 7 (b), the figure also indicates that when FPA score is less than 1, the correctness of the non-PI network is NOT always positively correlated with the FPA score.”*
>
> * Would you agree that having a method that can always tell whether a models’ prediction is incorrect without any label information is too good to be true? :)
> * **Our results serve to demonstrate a strong correlation** (much more than we initially expected) between per-instance accuracies and FPA scores. We’ve made sure we don’t claim perfect correlation.
>
> > _“Figure 7 (a) shows the residuals given by the two different solvers are different. However, this number is not rigorous enough to support the show that different solvers converge to different fixed points. It is better to directly compute the RELATIVE difference between the solutions given by two different solvers. Besides, to claim that this network is path independent, it will be more promising to show that its ATTACKED FPA score is close to 1, instead of the non-attached FPA score.”_
>
> * We have updated Table 1 to **include the results of adversarial stress testing** on this network i.e. an unrolled 32-layer weight-tied network with input injection trained with backprop. We find that this network is indeed path independent as indicated by its high attacked FPA score (0.9999) (computed on 500 random problem instances).
> * We have **added plots for per-instance analysis of test-time convergence in Appendix G**. We have also added plots that display relative differences between the solutions of the two different solvers. We find that there are instances where both the solvers clearly converge to different limiting behaviours on the same problem but can solve the problem successfully.

---

> > ### Author Response · Authors · 2022-08-08
> > **Thanks again for your review. Any other questions we could answer?**
> >
> > We’d like to thank you again for your review and thoughtful questions.
> >
> > With less than 24 hours left until the author-reviewer discussion ends, we wanted to ask if you have any remaining concerns that we could address.
> >
> > A particularly important improvement we’ve made during the discussion is:
> > * **New results using the Matrix Inversion task:** We also trained equilibrium models on the Matrix Inversion task introduced by Du et. al. [1] in their recently released (after NeurIPS deadline) paper. Equilibrium networks beat the baselines of [1] and matched the performance of their proposed method. We also saw the same trend we did on the other taks: lower degrees of path independence ended up with poorer generalization, with the best performing model having the (highest) FPA score of 1.
> >
> > We believe our improvements to the paper (several thanks to your questions! - please see our initial response) have made it significantly more rigorous and watertight. We hope that you would consider increasing your score if our response and paper updates sufficiently address your concerns. Thank you.

---

> ### Author Response · Authors · 2022-08-02
> **Response to Reviewer 4wU9 (1/2)**
>
> Thank you for your careful review and good questions. We believe we have solid answers to all of your concerns – we’ve updated the writeup accordingly. We hope that you would consider increasing your score if our response and paper updates sufficiently address your concerns.
>
> **Summary:**
> * **Clarification of FPA calculations**: Updated Algorithm Box 1 to clarify the procedure for calculation of Fixed point alignment (FPA) score.
> * **Extended analysis on per-instance test time convergence**: New experiments on per-instance analysis of test time convergence in Section G of appendix that clearly shows limit cycles.
> * **Additional results on adversarial stress testing**: Updated Table 1 to include results on prefix sum task
>
>
> **General Response**
>
> > *All the claims are only supported by experiments, without any theoretical justifications. The experiments are not rigorous enough.*
>
> Our submission is an empirical investigation (which we believe is valuable!) that highlights a number of highly nontrivial phenomena, and carefully collects evidence to back our claims up (listed below).
>   * Equilibrium models often learn path-independent functions, even when they’re not constrained to be so.
>     * **Carefully chosen metric**: We chose the FPA score through careful consideration. Please see Appendix A, which contains a detailed discussion of why FPA score is better than other reasonable alternatives. **Note that FPA score is unitless**. This allows direct comparisons between different trained networks. Note that naive approaches (like using Euclidean distance, or norms of sensitivity Jacobians) don’t make this possible.
>     * **Results based on hundreds of trained networks**: With a reliable metric at our hands, we trained hundreds of networks, empirically verifying that path independence is a pervasive phenomenon.
>   * Path independence strongly correlates with performance - especially on examples that need additional test-time compute.
>     * **Results robust against careful ablations**: We ablate key architectural and optimization hyperparameter to demonstrate this correlation.
>   * Interventions that promote/discourage path independence improve/hurt performance respectively.
>     * **Went beyond correlational analysis**: We were careful not to over-interpret our earlier correlational results, so we ran carefully picked interventional experiments to confirm path independence plays. This provides much stronger evidence than mere correlational analyses.

---

### Author Response · Authors · 2022-08-02
**General Response**

We thank the reviewers for their careful reviews and insightful questions. With the renewed interest in inference-time techniques in the NeurIPS community, we believe our findings make an **important conceptual contribution to our understanding of how certain systems, like equilibrium models, can utilize additional test-time compute.**

We’d like to highlight some improvements we’ve made to the paper:

* **Polished writing and plots:** We’ve significantly improved the writing and the clarify of the plots and algorithm boxes (please see the new Algorithm Box for computing Fixed Point Alignment Scores!).
* **New section explaining the rationale behind FPA scores:** We landed on FPA score to quantify path independence through a careful selection process. We outline these considerations in Appendix A. In summary, FPA score is reliable, efficient and unitless (allowing comparisons between trained networks) - qualities other methods we considered lack.
* **Additional results on BlurryMNIST:** We also show a positive correlation between path-independence and generalization on the BlurryMNIST dataset. This task involves training an image classifier on MNIST digits corrupted with small degrees of Gaussian blur, and testing the performance on significantly more corrupted ones.
* **Extensive per-instance convergence analysis:** We resolve lingering questions regarding training and test time convergence by running extensive experiments. This includes a definitive illustration of limit cycle behaviour.
* **Additional results on adversarial stress testing:** We extended the scope of our adversarial stress test on the FPA score to include additional results on prefix sum.

---

> ### Author Response · Authors · 2022-08-09
> **Thank you for following up**
>
> We thank the reviewers for following up.
>
> We’ve further improved our submission by adding new experiments, doing sanity checks and improving the writing. Most notable additions are 1) **new results on the Matrix Inversion task that confirm our findings** (this task is proposed by Du et. al. [1] which is released after the NeurIPS submission deadline) and 2) a sanity check experiment where **different ways of quantifying path independence all yield the same qualitative conclusions**.
>
> We hope that our improvements have answered the additional questions of the reviewers.
>
> [1] Du, Yilun, et al. "Learning Iterative Reasoning through Energy Minimization." International Conference on Machine Learning. PMLR, 2022.

---

### Meta-Review · Area_Chair_gcfg · 2022-08-26

**Recommendation:** Accept
**Confidence:** Less certain

**Metareview:**

This paper finds that there is a strong correlation between generalization to harder examples and path independence (PI), for equilibrium models. The paper proposes a simple-to-compute metric called the Fixed Point Alignment (FPA) score, to measure the level of PI.

During the rebuttal/revision phase, the paper was significantly improved, including clarifying Algorithm 1 and more thorough experimentation. Although some skepticism remains regarding the FPA score and the experiments, the reviewers agree that this paper has made clear and interesting contributions. Therefore it's worth having this paper presented at the conference.

**Award:**

No

---

### Decision · Program_Chairs · 2022-09-14

Accept